# Enhanced Heterogeneous Fenton-like Process for Sulfamethazine Removal via Dual-Reaction-Center Fe-Mo/rGO Catalyst

**DOI:** 10.3390/nano12234138

**Published:** 2022-11-23

**Authors:** Weihua Qin, Yueming Ma, Ting He, Jingbin Hu, Pan Gao, Shaoxia Yang

**Affiliations:** 1School of Water Resources and Hydropower Engineering, North China Electric Power University, Beijing 102206, China; 2National Engineering Laboratory for Biomass Power Generation Equipment, School of Renewable Energy, North China Electric Power University, Beijing 102206, China

**Keywords:** heterogeneous Fenton-like reaction, Fe-Mo dual-reaction-center catalyst, reactive oxygen species, sulfamethazine

## Abstract

A heterogeneous Fenton-like catalyst with single redox site has a rate-limiting step in oxidant activation, which limited its application in wastewater purification. To overcome this, a bimetallic doping strategy was designed to prepare a heterogeneous Fenton-like catalyst (Fe-Mo/rGO) with a double-reaction center. Combined with electrochemical impedance spectroscopy and density functional theory calculation, it was confirmed that the formation of an electron-rich Mo center and an electron-deficient Fe center through the constructed Fe-O-Mo and Mo-S-C bonding bridges induced a higher electron transfer capability in the Fe-Mo/rGO catalyst. The designed Fe-Mo/rGO catalyst exhibited excellent sulfamethazine (SMT) degradation efficiency in a broad pH range (4.8–8.4). The catalytic performance was hardly affected by inorganic anions (Cl^−^, SO_4_^2−^ and HCO_3_^−^) in the complicated and variable water environment. Compared to Fe/rGO and Mo/rGO catalysts, the SMT degradation efficiency increased by about 14.6 and 1.6 times in heterogeneous Fenton-like reaction over Fe-Mo/rGO catalyst. The electron spin resonance and radical scavenger experiments proved that ·O_2_^−^/HO_2_· and ^1^O_2_ dominate the SMT removal in the Fe-Mo/rGO/H_2_O_2_ system. Fe and Mo, as active centers co-supported on rGO, significantly enhanced the electron transfer between catalyst, oxidant, and pollutants, which accelerated the reactive oxygen species generation and effectively improved the SMT degradation. Our findings offer a novel perspective to enhance the performance of heterogeneous Fenton-like catalysts by accelerating the electron transfer rate in the degradation of organic pollutants.

## 1. Introduction

Sulfonamides, a kind of synthetic antibiotic, are widely used in animal husbandry to prevent and treat bacterial infectious diseases. It is reported that 60–90% of sulfonamides used in the world are discharged into the environment, which could seriously threaten the safety of aquatic ecosystems and human health [1]. Sulfamethazine (SMT), one of the sulfonamides, has caused greater concern due to its stable structure, which allows it to inhibit the biological treatment and high concentration exposed in the environment [2]. Recently, many technologies have been developed to destroy sulfonamides in wastewater. Adsorption is an effective technology to remove antibiotic pollutants in water, but it only achieved phase transfer and failed to achieve its purpose of removing antibiotics from the environment [3]. Advanced oxidation processes (AOPs) are gaining popularity as efficient treatment methods [4,5,6]. The homogeneous Fenton process, as one of the AOPs, is clean and efficient technology used to generate reactive oxygen species (ROS) to destroy hazardous organics through the reaction between Fe^2+^ and H_2_O_2_ under an acidic operating condition [7], while defects such as a narrow pH range and formation of iron sludge limit its development in the wastewater treatment. Compared with a homogeneous Fenton reaction, the heterogeneous Fenton-like system avoids these shortcomings. However, the reduction of M^n+^/M^(n+m)+^ on the solid catalysts, a rate-limiting step, also restrained the pollutant degradation efficiency [8].

To avoid these disadvantages, a novel dual-reaction-center mechanism was proposed to solve the rate-limiting step by generating non-uniform electron distribution on the surface of heterogeneous Fenton-like catalysts [9,10]. For example, Hu et al. prepared a dual-reaction-center CuTiAl-SiO_2_ nanocatalyst to destroy hazardous pollutants [7]. Stronger electron density was concentrated near the Cu atoms as the electron-rich centers, and weaker electron density was formed around Ti and Al as the electron-deficient centers. In the reaction, H_2_O_2_ was reduced to ROS on the electron-rich-center to oxidize the pollutants adsorbed on the catalyst. Electron transfer between the dual reaction centers was easily conducted due to electronegativity difference of these metal ions. Therefore, higher degradation efficiency of the pollutants was obtained over the dual-reaction-center heterogeneous catalysts [11].

In addition, the electron transfer capability in the heterogeneous catalysts would affect the ROS generation. The acceleration of electron transfer via chemical bonding bridges is an effective method to enhance the reactivity of heterogeneous catalytic process. For example, Zhuang et al. proposed that the formation of Fe-O-C bonding bridges in FeOOH/RGO catalyst accelerated the electron transfer [12]. Lyu et al. reported that a novel CMS-rGO nanocatalyst enhanced electron transfer by constructing Mo-O-Co chemical bonding bridges [13]. Some studies have proved that the Fe doping Mo-based catalyst significantly increased the electron transfer by the Fe-O-Mo bonding bridges [14,15,16]. Qu et al. had clearly corroborated that incorporating Fe-O-Mo electron mediator could accelerate the regeneration of Fe^2+^ to improve the Fenton-like reaction [17]. In addition, carbon materials with high specific surface area and higher electrical conductivity are helpful to the electron transfer from the adsorbed contaminant to active sites during the reactions [18,19]. Additionally, the carbon-supported Mo materials also exhibited good electron transfer due to the existence of the bonding bridges of Mo-C [15].

In this work, a novel Fe-Mo/rGO catalyst, with the dual-reaction center and high electron transfer capability, was prepared by a hydrothermal method, and SMT was selected as the objective pollutant. The catalytic performance of Fe-Mo/rGO catalyst was evaluated by the SMT degradation. Moreover, the mechanism of the Fe-Mo/rGO catalyst was also proposed to understand the SMT degradation over the Fe-Mo/rGO catalyst.

## 2. Materials and Methods

### 2.1. Materials

All chemical reagents, not further purified, were obtained from Aladdin Industrial Corporation (Shanghai, China). rGO was purchased from Suzhou Universal Technology Company (Jiangsu, China). rGO was calcined under N_2_ atmosphere at 350 °C for 3.5 h, and then used for the catalyst preparation.

### 2.2. Preparation of the Catalysts

The Fe-Mo/rGO catalyst was synthesized by a hydrothermal method. Then, 50 mg rGO and 220 mg (NH_4_)_2_MoS_4_ were added into 30 mL N,N-Dimethylformamide solution. The mixture was ultrasonically treated for 20 min to form a well-dispersed suspension. Next, 26.95 mg FeCl_2_·4H_2_O was added into the mixture under N_2_ bubbling, and the mixed solution was transferred to a high-pressure kettle kept for 10 h at 200 °C. After natural cooling, the obtained precursors were washed with deionized water to neutralize them, and dried in a vacuum oven at 60 °C overnight. The resulting solid was calcined at 350 °C for 1 h under N_2_ atmosphere to obtain the Fe-Mo/rGO catalyst. The Fe/rGO and Mo/rGO catalysts were prepared using the same method. Fe weight ratio was 23% and the molar of Fe/Mo was 1:4 in the Fe-Mo/rGO catalyst.

### 2.3. Material Characterization

X-ray diffraction (XRD, SmartLab SE, Rigaku, Toshima, Tokyo, Japan) was applied to characterize the crystal structure of the catalysts, using a Cu K_α_ target as an X-ray source (λ = 0.15406 nm) under 40 kV voltage in the angle range of 10°–80° along with a 6°/min scanning speed. The morphology of the catalysts was obtained via a scanning electron microscope (SEM, Quanta250, FEI, Hillsborough, OR, USA) equipped with a 200 kV energy dispersive spectrometer (EDS, EMAX-350, Horiba, Kyoto, Japan). X-ray photoelectron spectroscopy (XPS, AXIS ULTRA^DLD^, Shimadzu, Kyoto, Japan) was used to analyze the chemical sates of the catalyst surface elements. X-ray source was Al K_α_ (*hν* = 1486.60 eV) and X-ray working power was 150 W. The C 1s peak (*E_b_* = 284.60 eV) was used as a standard to calibrate the binding energy. Electrochemical impedance spectroscopy (EIS) was measured at open circuit voltage in the electrolyte solution of 5 mM K_3_[Fe(CN)_6_] + 0.1 M KCl, Hg/HgO as reference electrode, platinum wire as counter electrode, and glassy carbon electrode as working electrode. The density functional theory (DFT) calculation was performed in the Vienna ab initio simulation package (VASP). Spin-polarized GGA, PBE functional, and all-electron plane-wave basis sets had an energy cutoff of 420 eV. The convergence criteria for energy and force were set to 10^−5^ eV and 0.01 eV/Å, respectively.

### 2.4. Experimental Procedure

The degradation experiments of SMT were conducted in a 250 mL glass reactor. First, 50 mL SMT solution (0.076 mM) and 20 mg catalyst were mixed under vigorous stirring for 30 min, to reach the SMT adsorption equilibrium on the catalysts. Then, a certain amount of H_2_O_2_ was added to the mixed solution with stirring, and the reaction started. The solution was adjusted with a concentration of HCl (0.05 M) and NaOH (0.05 M) solution to obtain the desired initial pH value. Each experiment was performed three times. During the reaction, samples were taken at certain time intervals and quenched with methanol. The SMT was detected using high-performance liquid chromatography (HPLC, 1260, Agilent, Santa Clara, CA, USA) using C18 column under a detection wavelength of 254 nm. The mobile phase was a mixture of methanol and 0.2% acetic acid solution (40:60, *v*/*v*) with a flow rate of 1.0 mL/min. The metal ion leached during the reaction was measured using an atomic emission spectrometer (AES, ICPE-9820-Plasma, Shimadzu, Kyoto, Japan).

The quenching experiments were performed by adding tert-butanol (TBA), nitro blue tetrazolium (NBT), and L-histidine to detect ROS like hydroxyl radical (·OH), superoxide anion radical (·O_2_^−^/HO_2_·), and singlet oxygen (^1^O_2_), respectively. Electron spin resonance (ESR, JES-FA200, JOEL, Toshima, Tokyo, Japan), using the Bruker EMX 10/12 spectrometer, was applied to detect the ROS produced in the Fe-Mo/rGO/H_2_O_2_ system by adding 5,5-dimethyl-1-pyrrolidine N-oxide (DMPO) for ·OH or ·O_2_^−^/HO_2_· and 2,2,6,6-tetramethylpiperidine (TEMP) for ^1^O_2_ [20]. The measure was set as follows: center field of 3500 G, sweep width of 100 G, microwave frequency of 9.853 GHz, microwave power of 20 mW, modulation amplitude of 1G, and modulation frequency of 100 kHz.

### 2.5. Calculation Method

The pseudo-first-order kinetic model was applied to calculate the initial reaction rate constant for SMT degradation [6], and expressed by the following equation:(1)$$\ln(\frac{C_{t}}{C_{0}})=-k\cdot t$$
where *C_t_* was the concentration of SMT at instant *t*, *C*_0_ was the initial SMT concentration, *k* represented the initial reaction rate constant, and *t* was the reaction time.

The particle size of a crystal in Fe/rGO, Mo/rGO and Fe-Mo/rGO catalysts could be determined by estimating the crystallite size, which was calculated from the XRD spectra using the following Scherrer’s equation [15]:(2)$$d_{\mathrm{XRD}}=\frac{K\lambda }{B\cos\theta }$$
where d_XRD_ was the crystallite size, *K* was the Scherrer constant that was near to 1, *λ* was diffraction wavelength that was 1.5046 Å, *B* was corrected full width at half maximum, and *θ* was the diffraction angle.

## 3. Results and Discussion

### 3.1. Characterization of the Catalyst

The morphologies of the rGO and Fe-Mo/rGO samples were shown in Figure 1. rGO manifested the typical lamellar structure with smooth surface (in Figure 1a). In the SEM images of Fe-Mo/rGO catalyst, it still possessed a sheet-like structure with the metal nanoparticle uniform dispersion on the rGO sheets (in Figure 1c,d). Moreover, the element mappings of the Fe-Mo/rGO catalyst verified the existence and the good dispersion of Fe, Mo, O, and S elements (in Figure 1e). The results suggested that Fe and Mo were incorporated into the rGO nanosheets in the preparation process of the Fe-Mo/rGO catalyst.

XRD patterns of the different catalysts were shown in Figure 2. For rGO, the wide peak at 24° was designated as the characteristic plane of the graphite crystal structure. The peak of the rGO plane weakened in the Fe-Mo/rGO catalysts, indicating that the hybridization of Fe and Mo on the rGO plane resulted in a reduction in the expansion of the plane-to-layer spacing [21]. For the Fe/rGO catalyst, the Fe_3_O_4_ diffraction peaks at 35.5°, 43.1°, and 62.6° were observed (JCPDS19-0629) [22]. For the Mo/rGO and Fe-Mo/rGO catalysts, the peaks at 13.9°, 33.3°, 58.8°, and 67.1° matched well with MoS_2_ (JCPDS37-1492), and the peaks at 25.9°, 26.2°, 37.0°, 49.6°, and 53.6° were attributed to MoO_2_ (JCPDS01-0615) [23]. In the Fe-Mo/rGO catalyst, the diffraction peaks of the Fe_3_O_4_, MoS_2_, and MoO_2_ crystalline phases were observed, and their intensities of the relative Fe and Mo peaks decreased. According to the Scherrer’s equation, the calculated particle size of Fe_3_O_4_, MoS_2_, and MoO_2_ crystalline phases in Fe-Mo/rGO catalyst were 12, 7, and 24 nm, which were smaller than that in Fe/rGO (26 nm for Fe_3_O_4_) and Mo/rGO (10 nm for MoS_2_ and 50 nm for MoO_2_). The results indicated that the co-introduced Fe and Mo in rGO caused a smaller crystal size in the Fe-Mo/rGO than that in the Mo/rGO and Fe/rGO catalysts. Furthermore, the XRD spectra of the used Fe-Mo/rGO catalyst had no obvious change, which proved the good stability of Fe-Mo/rGO catalyst.

The XPS full scan spectrum showed that Fe, Mo, O, S, and C elements were successfully loaded in the Fe-Mo/rGO catalyst (in Figure 3a), in correspondence with the result of EDS mappings of the catalyst. In Figure 3b, the broad XPS spectrum of the catalyst showed two characteristic peaks at 710.5 and 725.5 eV were attributed to Fe 2p_3/2_ and Fe 2p_1/2_, respectively. The high-resolution Fe XPS peaks were fitted into four peaks, the peaks at 712.7 and 725.5 eV were assigned to Fe(III)-O, and the peaks at 710.5 and 723.1 eV were attributed to Fe(II)-O [18,24]. In Figure 3c, the peaks at 229.1 and 232.2 eV were attributed to Mo(IV)-S, and the peaks at 229.5 and 232.8 eV were corresponded to Mo(IV)-O in the catalyst. Additionally, the peak at 235.1 eV was characterized as Mo(VI)-O. Moreover, a narrow S 2s peak at 226.4 eV also appeared in the Fe-Mo/rGO catalyst. These results indicated that chemical bonds of Mo-S, Mo-O, and Fe-O were formed in the Fe-Mo/rGO catalyst, in agreement with that of XRD [13]. In Figure 3d, the C 1s peak could be assigned to three peaks. In addition to the characteristic peaks of rGO at 285.0 and 288.9eV, a strong binding energy peak at 285.7 eV was attributed to the C-S bond formed during rGO and MoS_2_ hybridization. According to the XPS and XRD results, the Fe-O-Mo and Mo-S-C bonding bridges were effectively formed in the Fe-Mo/rGO catalyst.

### 3.2. Catalytic Degradation of SMT

#### 3.2.1. Effect of Different Catalysts

Figure 4 revealed the SMT degradation efficiency over different catalysts near neutral pH value at room temperature (RT) in the Fenton-like reaction. H_2_O_2_ alone hardly degraded the SMT, and the rGO and Fe/rGO samples also showed less than 5% SMT removal in 30 min. The result indicated that H_2_O_2_ could not be effectively activated by rGO and Fe/rGO catalysts to remove SMT near neutral pH value. The Mo/rGO catalyst exhibited ca. 30% SMT removal in 30 min. Notably, as high as 78% SMT degradation efficiency was achieved over the Fe-Mo/rGO catalyst in 15 min. The pollutant degradation followed the pseudo-first kinetics over different catalysts (in Figure 4b). The apparent kinetic constant (*k*) of the Fe-Mo/rGO catalyst (0.2156 min^−1^) was ca. 13 and 96 times higher than that of the Mo/rGO and Fe/rGO catalysts, respectively. In addition, the Fe ion leaching of Fe-Mo/rGO catalyst was ca. 0.9 mg/L in the reaction, and almost no Mo ion leaching was detected. The homogenous Fe^2+^, as a catalyst, was investigated for the SMT degradation near neutral pH value in the presence of H_2_O_2_. It was found that ca. 10% SMT removal was obtained over Fe^2+^ ion in the reaction and was far lower than that over the Fe-Mo/rGO heterogamous catalyst. The above results demonstrated that the Fe-Mo/rGO catalyst had a better catalytic performance for the SMT degradation at near neutral pH value.

Generally, there were abundant inorganic anions in water and wastewater, which could react with ROS and affect the degradation of pollutants [14,25]. Therefore, the performance of the Fe-Mo/rGO catalyst for SMT removal near neutral pH value were evaluated in the presence of inorganic anions ([Cl^−^] = [SO_4_^2−^] = [HCO_3_^−^] = 0.076 mM). As reported previously, ·OH oxidized SO_4_^2−^ and Cl^−^ to generate other radicals with a lower oxidation potential, and HCO_3_^−^ quenched ·OH to produce non-reactive HCO_3_· [19], resulting in a decrease in the pollutant degradation. As shown in Figure 4c, a slight reduction in the SMT removal was observed in the presence of different inorganic anions, and the inhibition was in the order of Cl^−^ < SO_4_^2−^ < HCO_3_^−^. The result indicated that the performance of the Fe-Mo/rGO catalyst was not obviously interfered with by inorganic anions in the reaction.

To further evaluate the performance of the Fe-Mo/rGO catalyst, Figure 4d and Table 1 showed the SMT removals of different catalysts reported in the literature under mild reaction temperature (25–35 °C). For the Fe-Mo/rGO catalyst, the initial reaction rate, expressed as the SMT removal per gram of a catalyst per minute, was 13.92 mg_SMT_/(min·g_cat_) near neutral pH value. The initial reaction rate of the Fe-Mo/rGO catalyst was ca. 32 times higher than that of the Fe-Mn-SiO_2_ catalyst at similar operating conditions (pH = 6.5–7.0, 25 °C) [26]. Moreover, the initial reaction rates of the Fe_3_O_4_-Mn_3_O_4_, Fe_3_O_4_-Mn_3_O_4_/rGO, and Fe_3_O_4_/MWCNTs catalysts under the acidic operating conditions were significantly inferior to that of the Fe-Mo/rGO catalyst near neutral pH value [2,23,27]. This result proved that the Fe-Mo/rGO catalyst had a good catalytic performance for the SMT degradation near neutral pH value.

#### 3.2.2. Effect of the Operating Condition

The degradation efficiency of the pollutants in the heterogeneous Fenton-like reaction was commonly influenced by the operating conditions, involving catalyst dosage, H_2_O_2_ concentration, initial pH value, and pollutant concentration [2]. Figure 5a showed the influence of the Fe-Mo/rGO catalyst dosage on the SMT removal at 25 °C with an initial SMT concentration of 0.076 mM, initial H_2_O_2_ dosage of 20 mM, and pH value of 6.57. The SMT removal rapidly enhanced from 53 to 78% with the *k* value of SMT degradation from 0.0341 up to 0.2164 min^−1^, when the catalyst dosage increased from 0.1 to 0.2 g/L after 30 min reaction (in Figure 5a and Figure 6). The larger catalyst dosage resulted in the more exposed active sites on the catalysts, which was beneficial to the H_2_O_2_ decomposition to produce ROS for the SMT removal in the reaction. When the dosage of catalyst continuously increased to 0.3 g/L, a lower SMT degradation efficiency (ca. 70%) was obtained. This phenomenon might be due to the self-consumption of ROS generated by excessive catalysts, in agreement with previous reports [12].

The initial pH value was usually known as a sensitive factor that affected the pollutant removal [28]. The effect of initial pH value (4.84, 6.57 and 8.42) on the SMT degradation at 25 °C with an initial SMT concentration of 0.076 mM, catalyst dosage of 0.2 g/L, and initial H_2_O_2_ dosage of 20 mM. The results were illustrated in Figure 5b. The Fe-Mo/rGO catalyst showed excellent SMT removal in a wide pH range (4.84–8.42). Ca. 60% SMT degradation efficiency was attained at pH values of 4.84 and 8.42 in the Fe-Mo/rGO/H_2_O_2_ system. The *k* value were 0.0599, 0.2164, and 0.0518 min^−1^ with the increase in the initial pH value (in Figure 6b), and the SMT degradation efficiency peaked around neutral pH value.

As shown in Figure 5c, the effect of the H_2_O_2_ concentration (20, 30, and 40 mM) on the SMT removal was investigated over the Fe-Mo/rGO catalyst near neutral pH value at 25 °C, with an initial SMT concentration of 0.076 mM and catalyst dosage of 0.2 g/L. With the increase in H_2_O_2_ concentration, the SMT removal significantly decreased from 78 to 50%, and the *k* value also decreased from 0.2164 to 0.0446 min^−1^ (in Figure 6c). The reason for this phenomenon might be attributed to the reaction between the excess H_2_O_2_ and ROS, leading to ineffective consumption of ROS, consistent with the previous studies [2,12,23].

Figure 5d showed the effect of SMT concentration on its degradation efficiency over the Fe-Mo/rGO catalyst near neutral pH value at 25 °C with the catalyst dosage of 0.2 g/L and initial H_2_O_2_ dosage of 20 mM. Increasing the SMT concentration from 0.038 to 0.115 mM, a lower pollutant degradation was observed, and the *k* value decreased from 0.7879 to 0.1891 min^−1^ in the reaction over the Fe-Mo/rGO catalyst (in Figure 6d). Meanwhile, 100% SMT conversion was obtained after 5 min run at a lower SMT concentration (0.038 mM). This result may be due to the competitive adsorption of H_2_O_2_ and SMT that happened on the catalyst’s surface. The SMT with a high concentration occupied more active sites on the catalyst, blocked the H_2_O_2_ adsorption, and then restrained the ROS formation. The result was in agreement with the previous reports [2,8].

The effect of reaction temperature (10, 15, and 25 °C) on the SMT degradation efficiency was evaluated in the Fenton-like reaction over the Fe-Mo/rGO catalyst near neutral pH value (in Figure 7a). The SMT removal significantly increased from 32 to 78% as the reaction temperature increased from 10 to 25 °C. The apparent activation energy (*E*_a_) calculated by the Arrhenius equation was 30.9 kJ/mol in the heterogeneous Fe-Mo/rGO Fenton-like system (in Figure 7b). *E_a_* was lower than that of the N-RGO (*E*_a_ = 31.6 kJ/mol) and Fe_2_O_3_-CeO_2_ catalysts (*E*_a_ = 73 kJ/mol) for the SMT degradation [13,29,30], indicating that the prepared Fe-Mo/rGO catalyst in this study exhibited a better catalytic performance for the SMT degradation.

The stability of the Fe-Mo/rGO catalyst was evaluated in the Fenton-like reaction for the SMT degradation near neutral pH value by a five-run continuous test. As shown in Figure 8, the SMT removal gradually decreased in the recycle reaction, but it could be stable over 57% during the fifth cycle, which still maintained 72% of the original SMT removal efficiency. Therefore, the Fe-Mo/rGO catalyst exhibited good catalytic performance and stability for the SMT degradation. Generally, the dereased activity of catalysts were mainly attributed to two reasons, which resulted in the deactivation of the catalysts: one was the adsorption of intermediates on the catalyst surface [31]; another was the active metal leaching during the reaction [32]. In the study, a low metal iron leaching (ca. 0.10 mg/L) was found in the recycle reaction, which had little impact on the pollutants degradation in the reaction. Thus, the activity decrease of the Fe-Mo/rGO catalyst may be due to the carbon deposition caused by the adsorption of pollutants and intermediates on the surface of the catalyst during the reaction. In addition, our group’s previous study on the degradation of phenol in the Fenton-like reaction also showed a similar phenomenon. After the reaction, some of the surface pores in catalyst were blocked by the deposited carbonaceous intermediates [33], which again acts as evidence for the cause of the catalyst deactivation for contaminant degradation.

### 3.3. Reaction Mechanism

In order to determine ROS generated during the Fenton-like reaction for the SMT degradation over the Fe-Mo/rGO catalyst, the quenching experiments were conducted in this study. TBA, NBT, and L-histidine were used to detect widely produced ·OH, ·O_2_^−^/HO_2_·, and ^1^O_2_, respectively [12,34,35]. As shown in Figure 9a, it was found that (1) the SMT degradation efficiency significantly dropped to ca. 17% after the addition of NBT; (2) the SMT removal obviously decreased to ca. 36% with the introduction of L-histidine; (3) the SMT conversion had a slight decrease from 78% to 56% with the addition of TBA. In addition, the formed ROS in the Fe-Mo/rGO/H_2_O_2_ system were further determined by ESR. In Figure 9b, a weak signal of ·OH and two strong obvious signals of ·O_2_^−^/HO_2_· and ^1^O_2_ were observed on the ESR spectra. The intensity of ·O_2_^−^/HO_2_· signal was obviously higher than that of ^1^O_2_ and ·OH signals, which is inconsistent with the result of radical quenching experiments. These results indicated that ·O_2_^−^/HO_2_· and ^1^O_2_ played the decisive roles in the reaction for degrading SMT over the Fe-Mo/rGO catalyst [26]. In addition, the ESR signal intensities of generated ^1^O_2_ and ·O_2_^−^/HO_2_· were compared in the Fenton-like reaction for the SMT degradation over rGO, Fe/rGO, Mo/rGO, and Fe-Mo/rGO catalysts. In Figure 9c,d, it was found that ^1^O_2_ and ·O_2_^−^/HO_2_· signals were observed in the system over different catalysts, and their intensities were the same order, as follows: Fe-Mo/rGO > Mo/rGO > Fe/rGO > rGO. These data suggested that the Fe-Mo/rGO catalyst had better SMT degradation efficiency due to the higher yields of ROS.

EIS was used to investigate the electron transfer behavior of the above catalyst. The Nyquist plots were obtained for rGO, Fe/rGO, Mo/rGO, and Fe-Mo/rGO catalysts on an electrochemical workstation. As shown in Figure 10a, the Nyquist plot of the Fe-Mo/rGO and Mo/rGO catalysts was obviously smaller than that of rGO and the Fe/rGO catalysts. Moreover, the Fe-Mo/rGO catalyst showed the smallest semicircular arc, indicating that the Fe-Mo/rGO catalyst had the best electron transfer properties, which effectively accelerated the ROS formation and then improved the SMT degradation in the Fe-Mo/rGO/H_2_O_2_ system.

DFT calculations was applied to investigate the valence electron density distribution on the Fe-Mo/rGO catalyst. The optimized Fe-Mo/rGO model fragment geometry was shown in Figure 10b, and the corresponding 2D valence electron density coloring map was illustrated in Figure 10c,d. It was observed that Mo atom had larger electron distribution than Fe and O atoms in the Fe-O-Mo model fragment, confirming that the dual reaction centers, including the electron-rich Mo center and electron-poor Fe center, were formed on the surface of the Fe-Mo/rGO catalyst. At the same time, the existence of C supports could also provide electrons to the electron-rich Mo center through the Mo-S-C bonding bridge, thus further promoting the stronger polarization of the electron cloud density between the electron-rich Mo center and the electron-poor Fe center (in Figure 10d). Therefore, the increased electron cloud density between Fe and Mo was beneficial for the acceleration of the electron transfer in the Fe-Mo/rGO catalyst [36]. The result indicated that the better electron transfer capability of the Fe-Mo/rGO catalyst was achieved by the Fe-O-Mo and Mo-S-C bonding bridges, in good agreement with the EIS result.

On the basis of the above results, we proposed the possible reaction mechanism of SMT removal in the Fe-Mo/rGO/H_2_O_2_ system (in Figure 11). Due to the different electronegativity of metal active phases, the electron-rich Mo center and the electron-poor Fe center were formed in the Mo/rGO catalyst. The SMT was adsorbed on the electron-poor Fe center, and Fe captured electrons from SMT and transferred to the electron-rich Mo center through the Fe-O-Mo bonding bridge to maintain the electron supply in the electron-rich region. Moreover, SMT could also be adsorbed on the C support to donate electrons to the electron-rich Mo center through the Mo-S-C bonding bridge with a stronger polarization of electron density dispersion between the electron-poor Fe center and the electron-rich Mo center. In the Fe-Mo/rGO/H_2_O_2_ system, the adsorbed H_2_O_2_ accepted the electrons on the electron-rich Mo center to produce the ·OH and ·O_2_^−^/HO_2_· and to improve the ^1^O_2_ formation. Generally, the ROS could attack SMT via cleavage of the S-C bond or S-N bond to form 4,6-dimethylpyrimidin-2-amine, 4-(2-imino-4,6-dimethylpyrimidin-1(2H)-yl) aniline, and sulfanilic acid [37]. The above intermediates would break the C-N bond to generate 4-aminophenol, aniline, and phenol, and then further ring-open and oxidize into harmless small molecules through free-radical pathway [37,38]. During the reaction, the formation of Fe-O-Mo and Mo-S-C bonding bridges significantly accelerated the interfacial electron transfer rate and conquered the rate-limiting steps for the regeneration of active metal ions in traditional Fenton reaction. The result accelerated the formation of ROS in the heterogeneous Fe-Mo/rGO catalyst, thereby improving the SMT degradation efficiency.

## 4. Conclusions

In this study, the dual-reaction-center Fe-Mo/rGO catalyst was successfully prepared via the hydrothermal method. The Fe-Mo/rGO catalyst exhibited an enhanced electron transform capability due to Fe-O-Mo and Mo-S-C bonding bridges. This was beneficial to ROS formation and promoted the SMT degradation efficiency. The Fe-Mo/rGO catalyst showed a good degradation efficiency for SMT in a broad pH range (4.84–8.42), and its performance was not affected by inorganic anions (Cl^−^, SO_4_^2−^, HCO_3_^−^). Low activation energy (ca. 30.9 kJ/mol) was obtained for the SMT degradation over the Fe-Mo/rGO catalyst. The ESR and radical scavenger experiments proved that the major ROS required to degrade SMT were ·O_2_^−^/HO_2_· and ^1^O_2_ in the reaction. By accelerating the electron transfer rate of the dual-reaction-center heterogeneous catalysts, this strategy could effectively improve the degradation of pollutants in the Fenton-like reaction.

## Figures and Tables

**Figure 1 nanomaterials-12-04138-f001:**
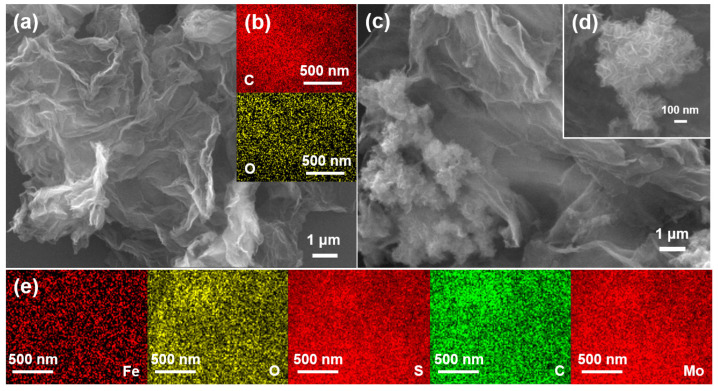
SEM spectra of catalysts of (**a**) rGO and (**c**,**d**) Fe-Mo/rGO; EDS elemental mappings of catalysts of (**b**) rGO and (**e**) Fe-Mo/rGO.

**Figure 2 nanomaterials-12-04138-f002:**
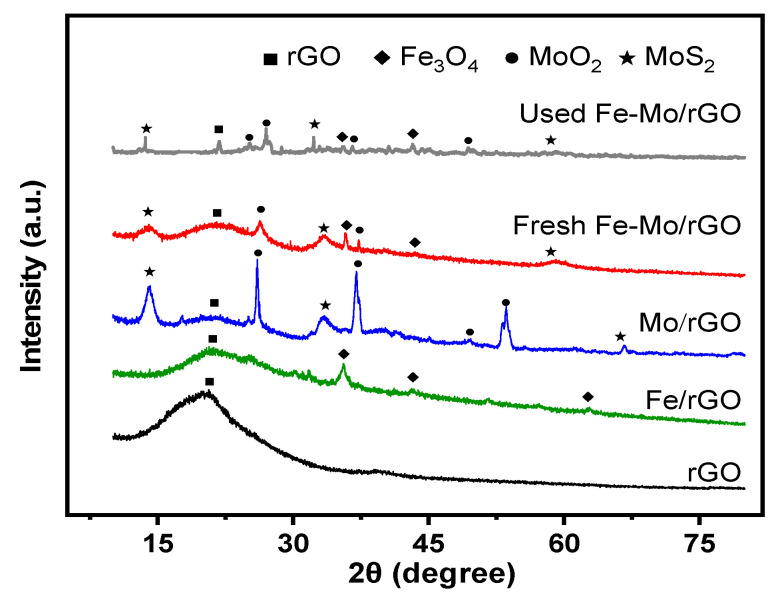
XRD patterns of different catalysts.

**Figure 3 nanomaterials-12-04138-f003:**
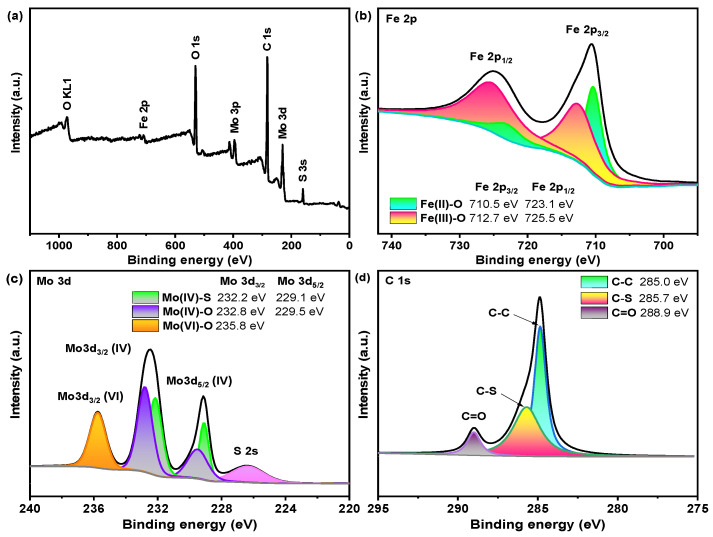
XPS spectra of Fe-Mo/rGO catalyst. ((**a**) full spectrum; (**b**) Fe 2p; (**c**) Mo 3d; (**d**) C 1s).

**Figure 4 nanomaterials-12-04138-f004:**
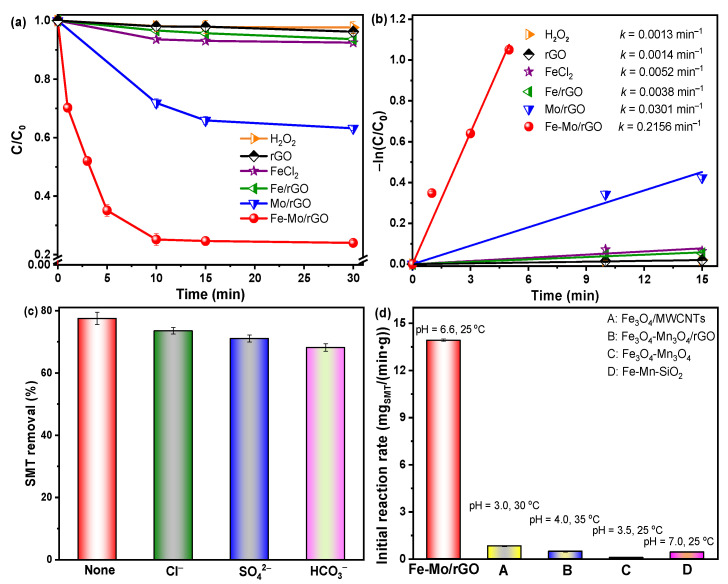
(**a**) The SMT removal and (**b**) the kinetics curves for the SMT removal in the Fenton-like reaction over different catalysts. (**c**) Effect of 0.076mM of different inorganic anions for SMT removal over Fe-Mo/rGO catalyst. (**d**) Comparison of the initial reaction rate for the SMT removal over different catalysts. ([SMT]_0_ = 0.076 mM, [H_2_O_2_]_0_ = 20 mM, pH = 6.57, T = 25 °C, [Catalyst] = 0.2 g/L).

**Figure 5 nanomaterials-12-04138-f005:**
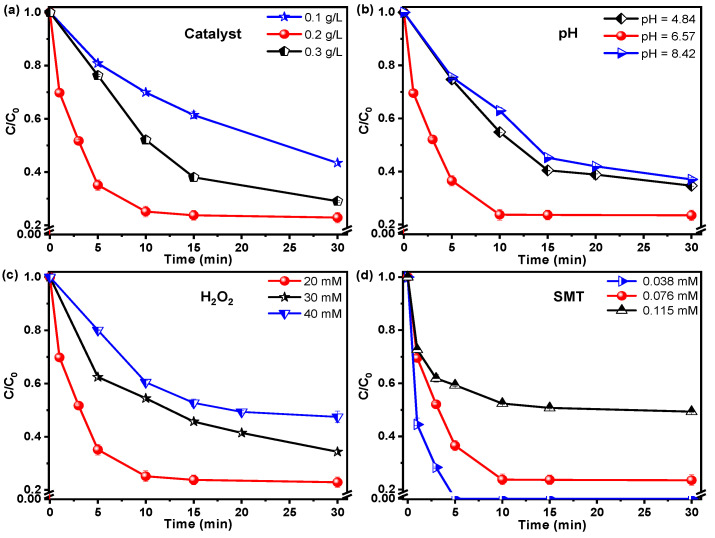
Effect of the operating conditions on the SMT removal over the Fe-Mo/rGO catalyst. ((**a**) Catalyst dosage: [SMT]_0_ = 0.076 mM, [H_2_O_2_]_0_ = 20 mM, pH = 6.57, T = 25 °C; (**b**) pH value: [SMT]_0_ = 0.076 mM, [Catalyst] = 0.2 g/L, [H_2_O_2_]_0_ = 20 mM, T = 25 °C; (**c**) H_2_O_2_ concentration: [SMT]_0_ = 0.076 mM, [Catalyst] = 0.2 g/L, pH = 6.57, T = 25 °C; (**d**) SMT concentration: [Catalyst] = 0.2 g/L, [H_2_O_2_]_0_ = 20 mM, pH = 6.57, T = 25 °C).

**Figure 6 nanomaterials-12-04138-f006:**
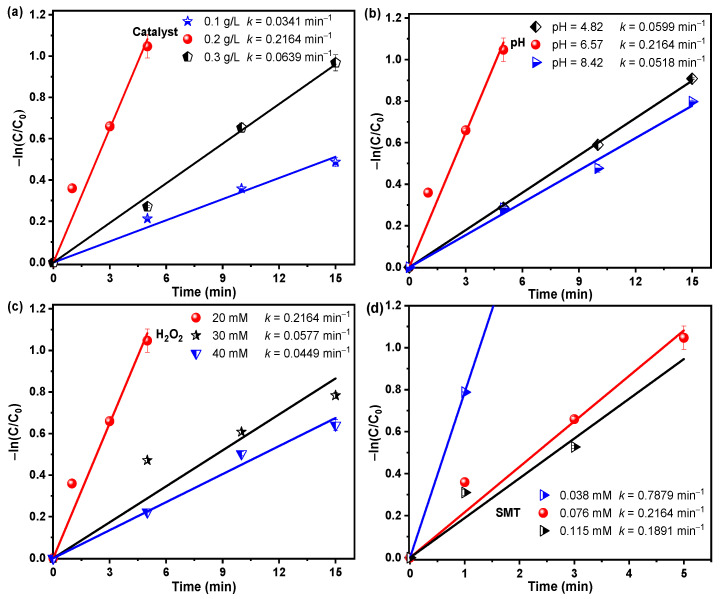
The kinetics curves for the SMT removal over the Fe-Mo/rGO catalyst under different operating conditions. ((**a**) Catalyst dosage: [SMT]_0_ = 0.076 mM, [H_2_O_2_]_0_ = 20 mM, pH = 6.57, T = 25 °C; (**b**) pH value: [SMT]_0_ = 0.076 mM, [Catalyst] = 0.2 g/L, [H_2_O_2_]_0_ = 20 mM, T = 25 °C; (**c**) H_2_O_2_ concentration: [SMT]_0_ = 0.076 mM, [Catalyst] = 0.2 g/L, pH = 6.57, T = 25 °C; (**d**) SMT concentration: [Catalyst] = 0.2 g/L, [H_2_O_2_]_0_ = 20 mM, pH = 6.57, T = 25 °C).

**Figure 7 nanomaterials-12-04138-f007:**
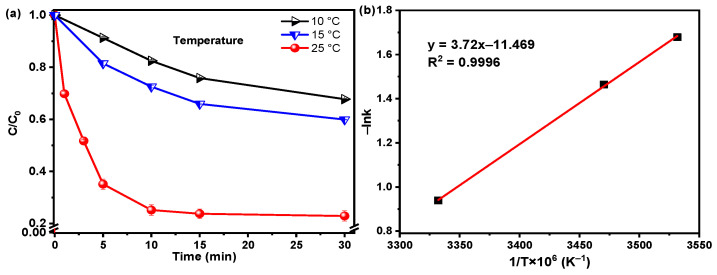
(**a**) Effect of the reaction temperature on the SMT removal and (**b**) the relation of 1/T and lnk in the Fenton-like reaction over the Fe-Mo/rGO catalyst. ([SMT]_0_ = 0.076 mM, [Catalyst] = 0.2 g/L, [H_2_O_2_]_0_ = 20 mM, pH = 6.57).

**Figure 8 nanomaterials-12-04138-f008:**
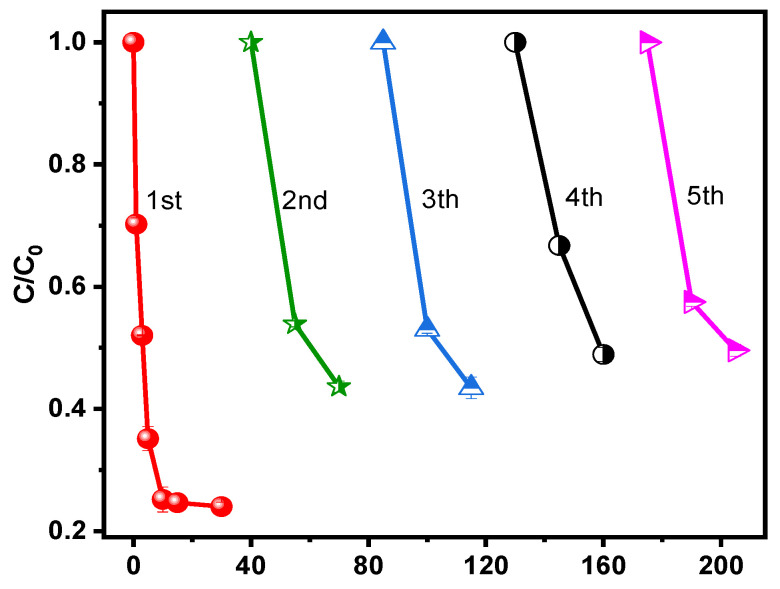
Reusability of the Fe-Mo/rGO catalyst for SMT removal. ([SMT]_0_ = 0.076 mM, [Catalyst] = 0.2 g/L, [H_2_O_2_]_0_ = 20 mM, pH = 6.57, T = 25 °C).

**Figure 9 nanomaterials-12-04138-f009:**
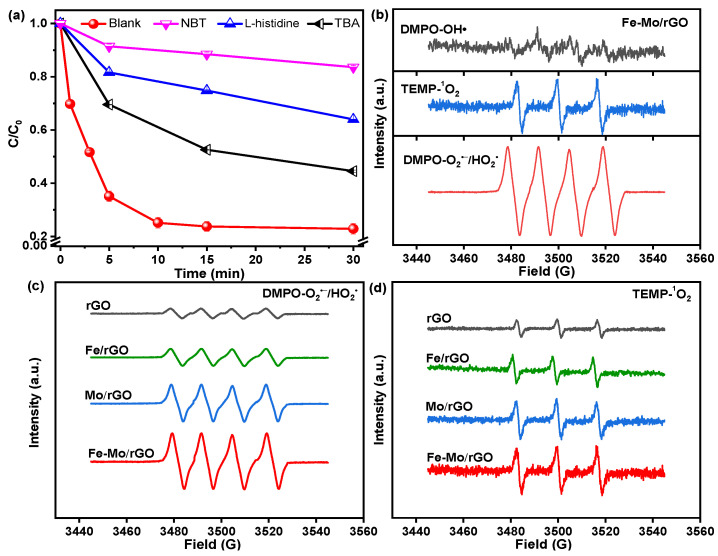
(**a**) The effect of radical scavengers on the SMT removal in the Fenton-like reaction over the Fe-Mo/rGO catalyst. (**b**) ESR spectra for detection of ROS in the heterogeneous Fe-Mo/rGO Fenton-like system. (**c**) ESR spectra for detection of ·O_2_^−^/HO_2_· in different systems. (**d**) ESR spectra for detection of ^1^O_2_ in the different systems.

**Figure 10 nanomaterials-12-04138-f010:**
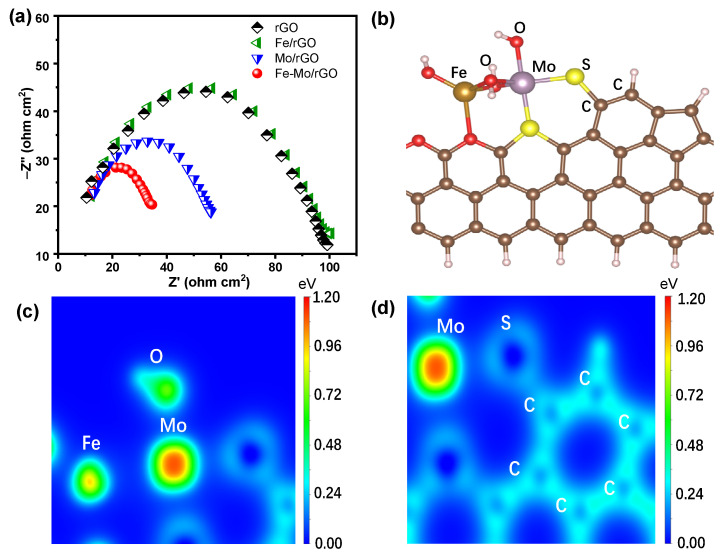
(**a**) EIS of different catalysts. (**b**) DFT calculations for the optimized structure of the Fe-Mo/rGO. The corresponding two-dimensional valence-electron density color-filled maps of the Fe-Mo/rGO model in (**c**) Fe-O-Mo vision fragment and (**d**) Mo-S-rGO vision fragment.

**Figure 11 nanomaterials-12-04138-f011:**
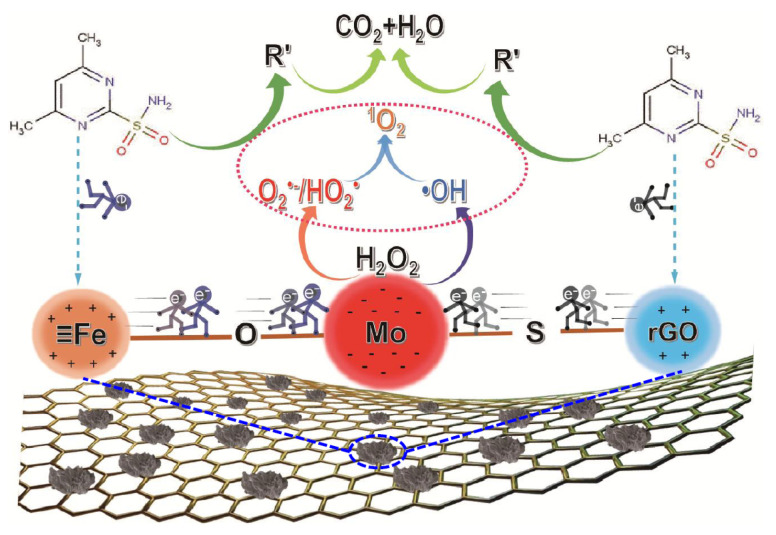
SMT degradation mechanism in the Fenton-like reaction over the Fe-Mo/rGO catalyst.

**Table 1 nanomaterials-12-04138-t001:** The SMT degradation over different iron-based catalysts in Fenton-like reaction.

Catalyst	C_0_ (mM)	Catalyst Dosage (g/L)	[H_2_O_2_]_0_ (mM)	pH	T (°C)	Reaction Time (min)	Removal Efficiency (%)	Initial Reaction Rate (mg_SMT_/min/g_cat_) ^1^	Reference
Fe-Mo/rGO	0.076	0.2	20.0	6.6	25	15	80	13.9	This work
Fe_3_O_4_-Mn_3_O_4_/rGO	0.07	0.5	6.0	3.0	35	80	98	0.5	[2]
Fe_3_O_4_-Mn_3_O_4_	0.07	0.5	6.0	3.0	45	50	100	0.05	[23]
Fe-Mn-SiO_2_	0.07	1.0	60.0	7.0	25	180	100	0.4	[26]
Fe_3_O_4_-MWCNTs	0.07	0.5	6.0	3.0	30	180	98	0.9	[27]

^1^ The SMT removal per gram of a catalyst per minute.

## Data Availability

Not applicable.

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
