# Peer review of "Enhanced Heterogeneous Fenton-like Process for Sulfamethazine Removal via Dual-Reaction-Center Fe-Mo/rGO Catalyst"

_nanomaterials, 2022, doi:10.3390/nano12234138_

Round 1

Reviewer 1 Report

 The authors reported the manuscript entitled "Enhanced heterogeneous Fenton-like process for sulfamethazine removal via dual-reaction-center Fe-Mo/rGO catalyst" nicely. Heterogeneous Fentons like catalysts play an important role in wastewater treatment. Paper is helpful to the scientific community. A heterogeneous Fenton-like catalyst (Fe-Mo/rGO)  has been synthesized and characterized very well. However, some points should be explained before acceptance:

1.     What is the calculated particle size and d- spacing of Fe-Mo/rGO catalyst from XRD spectra?

2.     Characterization of the catalyst (section 3.2) should be firstly described in the result and discussion (section 3) after that describe 3.1 Catalytic degradations of SMT part. Change this sequencing.

3.     Authors should provide the equations of the kinetic models used in this study.

4.     Add the following some recent papers references, that dealt with the adsorptive removal of antibiotics, such as

Nanomaterials 2021, 11, 568.

5. Novelty of the work should be clearly mentioned.  

6. In Line 22, the abstract section,fenon should be corrected as Fenton.

7. English language should be revised.

8. Compare the current data of your work to published data of related work.

Reviewer 2 Report

W. Qin et al. make a compelling case for using bimetallic Fe-Mo on rGO as dual-catalysts for sulfamethazine (SMT) degradation. The draft is well-written, with detailed mechanistic investigations. There are only a few observations to make before publication:

1. There are other bimetallic / dual-reaction-center catalysts, and some could be mentioned in the Introduction. To name a few (by DOI): https://doi.org/10.1038/s41598-019-52013-y; https://doi.org/10.1016/j.cej.2019.123324; https://pubs.acs.org/doi/10.1021/acscatal.6b00535; https://doi.org/10.3390/catal11010032; https://doi.org/10.1007/s10311-022-01453-6; https://onlinelibrary.wiley.com/doi/10.1002/eom2.12155.

2. When discussing the catalyst concentration, I suggest also to include the catalyst concentration vs that of SMT substrate, like in a typical catalytic reaction.

3. As a general observation, Fenton-like reaction should bear an abbreviation, as it's repeated many times, even in the same paragraph.

4. Figure 4, page 7: The units in Fig 4a: 100*c/c0 (since it's "%"); In fig 4b, on (Ox: it's not 1/T, but-from my calculations, 106 * 1/T. This must be corrected.

5. Lines 255-266: Has this hypothesis been checked? That C should have a lower thermal stability compared to rGO, or removed physically, to check if C is indeed the cause for performance degradation.

6. Figure 7: The XRD diffractogram also shows some unassigned peaks. Have the peaks in the range 35-60° (for used Fe-Mo/rGO) been assigned?

7. At XPS discussion: the valence of metal should be placed right next to it, i.e. Fe(III), Fe(II) etc

8. Lines 312-313: Is the Fe-S-Mo bonding a possibility?

9. Line 328: “in is consistent”

10. Line 331-332: rephrase.

11. Have the authors considered the partial decomposition of SMT -perhaps running an NMR on final solution, after evaporation and dissolution in , say, CDCl3? Like deprotection  of the SO2NH2 group, oxidation products etc.? The study is well written and interesting, but this piece of information would make the overall picture complete.

Round 2

Reviewer 1 Report

The authors revised the manuscript in light of the referee's comments very well. The authors have responded to all the queries raised by the referees.

Paper may be accepted in its current form.